# MiR-126 Is an Independent Predictor of Long-Term All-Cause Mortality in Patients with Type 2 Diabetes Mellitus

**DOI:** 10.3390/jcm10112371

**Published:** 2021-05-28

**Authors:** Justyna Pordzik, Ceren Eyileten-Postuła, Daniel Jakubik, Pamela Czajka, Anna Nowak, Salvatore De Rosa, Aleksandra Gąsecka, Agnieszka Cieślicka-Kapłon, Piotr Sulikowski, Krzysztof J. Filipiak, Dagmara Mirowska-Guzel, Jolanta M. Siller-Matula, Marek Postuła

**Affiliations:** 1Department of Experimental and Clinical Pharmacology, Centre for Preclinical Research and Technology (CEPT), Medical University of Warsaw, Banacha 1B, 02-097 Warsaw, Poland; justyna.pordzik@ouh.nhs.uk (J.P.); cereneyileten@gmail.com (C.E.-P.); dr.jakubik@gmail.com (D.J.); czajka.pamela@gmail.com (P.C.); nowania.a@gmail.com (A.N.); dmirowska@wum.edu.pl (D.M.-G.); jolanta.siller-matula@meduniwien.ac.at (J.M.S.-M.); 2Division of Cardiology, Department of Medical and Surgical Sciences, “Magna Graecia” University, 88100 Catanzaro, Italy; saderosa@unicz.it; 3Department of Cardiology, Medical University of Warsaw, 02-097 Warsaw, Poland; aleksandra.gasecka@wum.edu.pl (A.G.); agnieszka.kaplon@gmail.com (A.C.-K.); krzysztof.filipiak@wum.edu.pl (K.J.F.); 4Department of Computer Science and Information Technology, West Pomeranian University of Technology, 71-210 Szczecin, Poland; psulikowski@wi.zut.edu.pl; 5Department of Internal Medicine II, Division of Cardiology, Medical University of Vienna, 1090 Vienna, Austria

**Keywords:** microRNA, cardiovascular disease, diabetes, mortality, antiplatelet therapy, platelet reactivity, Let-7e, miR-126, miR-223, miR-125a-3p

## Abstract

MicroRNAs are endogenous non-coding RNAs that are involved in numerous biological processes through regulation of gene expression. The aim of our study was to determine the ability of several miRNAs to predict mortality and response to antiplatelet treatment among T2DM patients. Two hundred fifty-two patients with diabetes were enrolled in the study. Among the patients included, 26 (10.3%) patients died within a median observation time of 5.9 years. The patients were receiving either acetylsalicylic acid (ASA) 75 mg (65%), ASA 150 mg (15%) or clopidogrel (19%). Plasma miR-126, miR-223, miR-125a-3p and Let-7e expressions were assessed by quantitative real time PCR and compared between the patients who survived and those who died. Adjusted Cox-regression analysis was used for prediction of mortality. Differential miRNA expression due to different antiplatelet treatment was analyzed. After including all miRNAs into one multivariate Cox regression model, only miR-126 was predictive of future occurrence of long-term all-cause death (HR = 5.82, 95% CI: 1.3–24.9; *p* = 0.024). Furthermore, miR-126, Let-7e and miR-223 expressions in the clopidogrel group were significantly higher than in the ASA group (*p* = 0.014; *p* = 0.013; *p* = 0.028, respectively). To conclude, miR-126 expression is a strong and independent predictor of long-term all-cause mortality among patients with T2DM. Moreover, miR-223, miR-126 and Let-7e present significant interactions with antiplatelet treatment regimens and clinical outcomes.

## 1. Introduction

Type 2 diabetes mellitus (T2DM) constitutes the most prevalent chronic metabolic disorder, and its prevalence is progressively increasing. Notably, mortality rates in T2DM patients are known to be up to four-times as high in contrast to subjects without T2DM and predominantly associated with an increased hazard of cardiovascular disease (CVD) [1].

In the context of T2DM, platelets emerge as triggers of vascular injury induced by metabolic disturbances. The hypercoagulable state, which is often observed in T2DM, stems from disrupted platelet activation, aggregation, alterations in the coagulation processes, as well as endothelial dysfunction due to dysregulation of several signaling pathways [2]. In line with our previous reports, several genetic polymorphisms may contribute to differential platelet reactivity and outcomes in T2DM patients on antiplatelet treatment [3,4,5]. As we observe, platelets play a central role in the intertwined pathology linking T2DM, progression of atherosclerosis and occurrence of cardiovascular (CV) complications [6,7,8,9]. Since platelets are involved in atherogenesis and its complications, the blockade of one or numerous pathways regulating platelet activation and aggregation processes is essential in diminishing atherothrombotic risk in T2DM subjects.

While various mechanisms mediating CVD in diabetes have been determined, the need for identification of biomarkers is as important as ever. Over the last few years, platelets have been found to be a major source of miRNAs, which are highly conserved endogenous, non-coding RNAs that regulate gene expression at a post-transcriptional level through targeting messenger RNAs (mRNAs) [10,11]. Although miRNAs are found mostly intracellularly, a substantial amount is present in the extracellular space, such as the blood [12]. The extracellular miRNAs, which function as chemical messengers and mediators of cell-cell communication, offer exciting opportunities to become biomarkers [13]. Studies showed that platelets release specific miRNAs that can be predictive factors of morbidity and mortality across various diseases, such as coronary artery disease (CAD) and heart failure (HF) and provide information on platelet function itself [14,15]. Recently, miRNAs have been suggested as a potential tool to predict and monitor disease progression and therapeutic effectiveness. Several studies showed that miRNA expression can be regulated by antiplatelet treatment often utilized in higher risk populations [16,17,18,19]. Hence, miRNA signature profiles could be used to help modify antiplatelet treatment for the best prognostic outcomes in this cohort.

We aimed to determine the prognostic performance of selected circulating miRNAs for all-cause mortality and their potential usefulness as biomarkers in T2DM patients on antiplatelet treatment. Several studies showed the role of miR-126 and miR-223 in the context of antiplatelet treatment in many different cohorts [16,19,20,21,22,23]. Changes in plasma miRNAs in response to antiplatelet therapy were investigated in a small cohort of healthy volunteers [16,19,20,21,22,23]. Yu et al. analyzed the correlation between plasma miR-126 and miR-223, among others, and the risk of major adverse cardiovascular events in patients on dual antiplatelet therapy (DAPT) after percutaneous coronary intervention [16,19,20,21,22,23]. The association between miRNAs and platelet reactivity was also studied in patients with acute coronary syndrome (ACS) on various antiplatelet therapies [16,19,20,21,22,23]. In a similar study Carino et al. assessed miR-126 and miR-223, among other miRNAs, in patients with ACS before and after the therapeutic switch from DAPT to ticagrelor [16,19,20,21,22,23]. More recently the association of platelet-related miRNAs was studied in ACS patients on DAPT and pathophysiological processes in the TDM population [16,19,20,21,22,23]. Among others, Let-7e and miR-125a-3p are molecules that have been formerly linked to inflammation, platelet reactivity, and mechanisms leading to diabetes; however, the impact of antiplatelet therapy on Let-7e and miR-125a-3p and their prognostic potential has not been studied so far [24,25,26]. Therefore, our aim was to analyze the impact of antiplatelet treatment on Let-7e and miR-125a-3p expressions and confirm the previous observations for miR-126 and miR-223 expressions, as well as investigate their usefulness as predictive biomarkers specifically in the T2DM cohort.

## 2. Materials and Methods

### 2.1. Study Design

The ethics committee of the Medical University of Warsaw approved both the study protocol and the informed consent form. The study was conducted in accordance with the current version of the Declaration of Helsinki at the time when the study was designed, and informed written consent was obtained. Consecutive patients with T2DM presented at the outpatient clinic of the Central Teaching Hospital of the Medical University of Warsaw and recruited into the multi-center, prospective, randomized and open-label AVOCADO (Aspirin vs./Or Clopidogrel in Aspirin resistant Diabetes inflammation Outcomes) study were included in the present analysis. The full characterization of the study population, including the inclusion and exclusion criteria, was published previously [5]. Shortly, the AVOCADO study included patients aged between 30 and 80 years, with T2DM, irrespective of the type of antidiabetic treatment (with the exception of patients treated only with diet), burdened with at least two additional cardiovascular (CV) risk factors and receiving 75 mg of acetylsalicylic acid (ASA) daily. All inclusion criteria for the study included: age from 30 to 80 years of age, type 2 diabetes diagnosed at least 6 months before recruitment to the study requiring treatment with oral hypoglycemic drugs and/or insulin, history of cardiovascular events (i.e., myocardial infarction or ischemic stroke) or high risk of cardiovascular events defined as presence of at least 2 of the cardiovascular risk factors listed below: overweight or obesity diagnosed on the basis of the BMI (Body Mass Index) ≥ 25 kg/m^2^ (overweight), ≥ 30 kg/m^2^ (obesity), history of hypertension (SBP > 130 mmHg, DBP > 80 mmHg or pharmacological treatment), history of dyslipidemia (total cholesterol > 175 mg/dL, HDL-cholesterol: men < 40 mg/dL, women < 50 mg/dL, LDL-cholesterol > 100 mg/dL, or > 70 mg/dL (with concomitant disease ischemic heart disease), triglycerides > 150 mg/dL or pharmacological treatment); documented coronary artery disease (diagnosis previously made on the basis of objective clinical studies), documented cerebral vascular disease (diagnosis previously based on objective clinical studies), documented peripheral vascular disease (diagnosis previously made on the basis of objective clinical studies or a clinical event requiring hospitalization), positive family history of cardiovascular diseases and sudden cardiac death at an early age (female relatives of the 1st degree <55 years of age) 1st degree male relatives <65 years of age), history of smoking [5]. The exclusion criteria included: bleeding diathesis; history of gastrointestinal bleeding; platelet count < 150,000 per mm^3^; hemoglobin concentration < 10 g/dL; hematocrit < 30%; end-stage chronic renal disease requiring dialysis; anticoagulants (i.e., low-molecular—weight heparin, warfarin or acenocoumarol) or an alternative antiplatelet therapy other than ASA (i.e., ticlopidine, clopidogrel, prasugrel or dipyridamole); self-reported use of non-prescription non-steroidal anti-inflammatory drugs or drugs containing ASA within 10 days of enrollment; a recent history (within 12 months) of MI, unstable angina, coronary angioplasty or coronary artery bypass grafting; and a major surgical procedure within the previous 8 weeks [5]. The aim of the AVOCADO trial was to assess the effect of an eight-week course of clopidogrel or an increased ASA dose (150 mg) in patients with T2DM and high platelet reactivity on lower ASA dose (75 mg) as previously described [5]. Patients who did not exhibit high platelet reactivity on ASA 75 mg daily continued therapy, and patients with high platelet reactivity were randomized to ASA 150 mg or clopidogrel 75 mg in 1:2 ratio. Briefly, 197 patients were taking 75 mg ASA, 41 patients were taking 150 mg ASA, and 58 patients were taking clopidogrel. For the current analysis, we used only available blood samples taken 8 weeks after randomization according to the initial study design. Briefly, blood samples were taken in the morning 2–3 h after ingestion of antiplatelet drugs, as previously described in more detail [3,5,27,28,29]. Blood was kept at room temperature for 30 min before centrifugation at 1500 *g* for 15 min at 18–25 °C. Plasma was pipetted into 500 μL aliquots on ice and transferred to a −80 °C freezer for storage. For the purpose of this study, existing samples from patients already randomized to antiplatelet therapy based on study protocol were used, i.e., up to 6 weeks on therapy based on platelet reactivity measured at baseline, as previously described [5,28].

### 2.2. Study Endpoints

The primary endpoint was defined as all-cause death during the follow-up. The secondary endpoint was a composite of death, myocardial infarction (MI), unstable angina, and stroke or transient ischemic attack (TIA) at follow-up [30]. The composite endpoint was defined in accordance with the current universal criteria [31,32].

### 2.3. RNA Preparation and Detection and Quantification of miRNAs by Quantitative PCR

Plasma RNA was extracted by miVANA PARIS Kit. Total RNA was obtained as outlined above and diluted 1:10. Diluted RNA (5 μL) was reverse transcribed using the TaqMan miRNA Reverse Transcription kit (ABI) according to the instructions of the manufacturer. Subsequently, 3 μL of the product was used for detecting miRNA expression by quantitative polymerase chain reaction using TaqMan miRNA Assay kits (ABI) for the corresponding miRs on a The CFX384 Touch Real-Time PCR Detection System (BioRad Inc., Hercules, CA, USA). cel-miR-39 was added as a spike-in control. Reactions were run in triplicate, and the mean value was used for all analyses, to control variability associated with methodological reasons [33,34].

### 2.4. Statistical Analysis

Risk factors, clinical data and categorical variables are presented as percentages of patients and were compared using χ2 or Fisher’s exact tests, as appropriate. Continuous data are expressed as median and interquartile range and compared using Mann–Whitney U test for two independent samples. The distribution of data was checked with the Kolmogorov–Smirnov test. The Kaplan–Meier method was utilized for construction of survival curves. The log-rank test was applied to evaluate differences between groups. Proportional Cox-regression analysis was used to adjust for confounding factors. The following variables (age, gender, history of previous stroke, history of smoking, eGFR < 30) were entered into the Cox model on the basis of known clinical relevance or significant association observed at univariate analysis. We performed Cox regression analyses for each single miRNA and the clinical variables and also a final model in which all four miRNA were included in addition to the clinical variables. Effect estimates were presented as adj. hazard ratios (HR) and 95% CI. All tests were two-sided, and *p*-value < 0.05 was considered statistically significant. Calculations were performed using SPSS version 22.0 (IBM Corporation, Chicago, IL, USA). Based on 19% long-term mortality in the high-level miR-126 group as compared to 4% in the low-level miR-126 group, we calculated that with at least 125 patients per miR-126 group (1:1 sampling ratio, overall n = 250), our analysis had 96% power to detect differences in the risk of mortality between the miR-126 groups with a two-sided alpha value of <0.05.

## 3. Results

### 3.1. Patient Demographics

Patient demographics, T2DM data, concomitant diseases, medication and laboratory results are summarized in Table 1 and Appendix A. Out of 303 patients included, blood samples for analyses presented in this study were unavailable for 51 individuals. In fact, 48 patients did not present within the given time window for blood sampling, and in 3 remaining patients, the available blood volume was insufficient for miRNAs measurements. Among the 252 patients included, 26 (10.3%) patients died within a median observation time of 71 months (5.9 years). CV risk factors, such as hypertension (92%), dyslipidemia (85%) history of smoking (57%) and CAD (55%) were common in the majority of patients. Patient characteristics according to selected miRNAs are included in Appendix A.

### 3.2. Circulating miRNA Levels Predict Long-Term All-Cause Death

Patients who died had a 7.5-, 3- and 14-fold higher expression of miR-126, Let-7e and miR-125a-3p as compared with patients who survived, respectively (*p* < 0.001 for all miRNAs; Figure 1a,b,d). The difference in the expression for miR-223 did not differ statistically between those who died or survived (*p* = 0.089; Figure 1c). Since RNA was extracted from blood plasma for the purpose of this investigation, we refer to the expression of miRNA from extracellular space across this manuscript.

### 3.3. Predictive Value of miR-126, Let-7e, miR-223 and miR-125a-3p of Long-Term All-Cause Mortality

The study population was divided into two subgroups by using ROC curve analysis for each miRNA, i.e., low- or high value of single miRNAs (see Table 2 and Figure 2). The cut-off value of ≥2.08, labelled as high miR-126 level (40.5% of the population), the cut-off value of ≥0.82, labelled as high Let-7e level (47.6% of the population), the cut-off value of ≥6.62, labelled as high miR-223 level (54% of the population) and the cut-off value of ≥0.0017, labelled as high miR-125a-3p level (40.9% of the population) provided the prediction of all-cause mortality.

### 3.4. Expression of miRNAs According to the Allocation to the Antiplatelet Treatment Strategy

The expression of miR-126, Let-7e and miR-223 was significantly higher in the clopidogrel group as compared to the ASA 75 mg group (*p* = 0.015; *p* = 0.014; *p* = 0.024, respectively). This difference was not seen when comparing clopidogrel and ASA 150 mg groups for miR-125a-3p (Figure 3a–c; Appendix A). However, when comparing miRNAs expression between clopidogrel and the whole ASA groups (i.e., 75 mg +150 mg), miR-126, Let-7e and miR-223 expressions remained significantly higher in the clopidogrel subgroup (*p* = 0.014; *p* = 0.013; *p* = 0.028, respectively) (Figure 4).

### 3.5. Survival Analysis According to miRNAs Expression

The primary endpoint occurred in 26 (10.3%) out of 252 patients, for whom miR-126, Let-7e, miR-223 and miR-125a-3p levels were available. Six of the patients who died (4%) had low miR-126 values, and 20 patients (19%) had high miR-126 values, whereas for Let-7e, 5 patients (4%) had low Let-7e values, and 21 patients (17.5%) had high Let-7e values. Moreover, six of these patients who died (5%) had low miR-223 values, and 20 patients (14.5%) had high miR-223 values, whereas for miR-125a-3p, 9 patients (6%) had low miR-125a-3p values, and 17 patients (17%) had high miR-125a-3p values.

The primary endpoint in long-term follow-up was significantly more common in patients with high miRNA levels vs. low miRNA levels, for all analyzed miRNAs, i.e., miR-126, Let-7e, miR-223 and miR-125a-3p (*p* < 0.001; *p* < 0.001; *p* = 0.016; *p* = 0.005, respectively).

Adjusted time to event analyses have shown that heightened expression of miRNAs was associated with survival, when the models included one single miRNA and other covariates: miR-126 (HR = 7.31, 95% CI: 2.63–20.28; *p* < 0.001), Let-7e: (HR = 5.85, 95% CI: 2.08–16.46; *p* = 0.001), miR-223: (HR = 3.07, 95% CI: 1.17–8.07; *p* = 0.023), miR-125a-3p (HR = 2.93, 95% CI: 1.27–6.83; *p* = 0.013) (Figure 5; Table 3 and Appendix A). After inclusion of all 4 miRNAs into one multivariate Cox regression model, increased expression of miR-126 was associated with a 5.8-fold higher risk for long-term all-cause mortality (HR = 5.8, 95% CI: 1.2–24.9; *p* = 0.024), whereas the other miRNAs did not reach the statistical significance. Additionally, age (HR = 1.1, 95% CI: 1.01–1.12; *p* = 0.009) and male gender (HR = 4.06; 95% CI: 1.24–13.34; *p* = 0.027) were found independently associated with long-term all-cause mortality (Table 4).

## 4. Discussion

The present study determined the association of miR-126 with long-term all-cause mortality and antiplatelet treatment in patients with T2DM. It demonstrates two major findings. Namely, we revealed that high expression of plasma miR-126 may independently predict the risk of long-term all-cause mortality in the T2DM population. We also found that patients on clopidogrel treatment had higher miR-126, Let-7e and miR-223 expression compared to patients treated with ASA. It can be concluded with some caution that clopidogrel may be less beneficial than ASA 150 mg in T2DM with increased platelet reactivity; however, further studies on larger cohorts should be designed to elucidate the underlying mechanisms of this phenomenon. In a recent report by Angiolillo et al., DM was deemed to be one of the risk factors for high platelet reactivity, clopidogrel non-responsiveness and subsequent increased risk for adverse ischemic events [35].

MiR-126 belongs to the most abundantly expressed miRNAs in endothelial cells and is responsible for vascular development, integrity and response to hemodynamic stress [36]. miRNAs (miR-1, miR-133a and miR-19a) were included into one regression model; miR-126 and miR-223 were not predictive of death [37]. Moreover, Schulte et al. did not find a significant predictive value of miR-126 in CV mortality in CAD patients. Nevertheless, they reported that high miR-223 was predictive of future CV mortality, with 2.1% of the investigated cohort experiencing CV death over a median follow-up time of 4 years [38]. In our analysis, we showed that patients with high miR-126 expression had a 5.8-fold increased risk to experience death from any reason in long term follow-up. The exact mechanism of miR-126 is yet to be fully elucidated; however, its role in diabetes and vascular inflammation was reported. Even though miR-126 has been linked to angiogenesis and to the development of CVD, its role in platelet activation was also demonstrated [36,39,40]. Inconsistent findings have been reported on the predictive potential of miR-126 in high-risk populations, including patients with diabetes. Previous studies showed a correlation between low miR-126 and miR-223 expressions and increased all-cause and CV mortality, principally in patients with overt HF, as well as in individuals with diabetes. Witkowski et al. demonstrated that vascular tissue factor (TF), which is prompted by hyperglycemia and triggers pro-thrombotic conditions in diabetes, is controlled on the post-transcriptional level by miR-126, among others [41,42]. As reported by the team, coexpression of miR-126 with miR-19a results in control of vascular inflammation and amplifies the post-transcriptional regulation of vascular TF by exhibiting a cooperative suppression of the TF transcript in a luciferase reporter assay. However, it is worth noticing that the study was performed in a relatively small cohort of 44 patients with diabetes who received different hypoglycemic agents, and hence, the results could be biased [41].

The let-7 family are amongst the most highly expressed miRNAs in platelets and are actively secreted in microvesicles [43]. Moreover, Let-7e has a potential role in platelet reactivity, platelet regulatory pathways and apoptosis and is linked to inflammatory response in vascular endothelial cells [44,45]. To date, the influence of Let-7e on antiplatelet treatment has not been studied. In the present analysis, we demonstrated for the first time that Let-7e expression can be altered by different antiplatelet treatment regimens. Let-7e expression was found to be lower in patients taking ASA compared to clopidogrel group. This difference can be due to the anti-inflammatory effect of ASA. Importantly, Let-7e has a significant predictive power and correlates significantly with platelet function itself. A further possible explanation is the fact that antiplatelet treatment itself may exert an impact on circulating levels of miR-223 and miR-126, thus attenuating their prognostic value [46]. Previous studies reported modulation of circulating miRNAs in response to different antiplatelet treatments and in relation to platelet function. Nevertheless, varying—sometimes contrasting—results have been reported in multiple studies [16,21,23,47]. In our study, despite different doses of ASA, no significant impact was observed on circulating levels of miR-126, miR-223, miR-125a-3p and Let-7, whereas a significant difference in the expression of miR-126, miR-223 and Let-7 was observed in patients treated with the purinergic receptor (P2Y_12_) antagonist clopidogrel. The scarce data should be supported by further studies on larger cohorts in order to explain the influence of antiplatelet treatment on miRNAs expression.

In line with our results, miR-223 expression was shown to be differentially expressed in patients on antiplatelet treatment owing to the presence of binding site for miR-223 in natural 3′UTR in P2Y_12_ mRNA. This finding supports the hypothesis that P2Y_12_ expression could be modulated by miR-223 in platelets [17]. Reduced levels of plasma miR-223, primarily of platelet origin, were suggested to be a marker of efficacy of antiplatelet therapy [16,17,21,23,48,49]. However, Chyrchel et al. reported that decreased plasma miR-223 is not a marker of platelet responsiveness to dual antiplatelet therapy (DAPT). Instead, more potent platelet inhibition linked to newer P2Y_12_ antagonists seems to correspond with higher miR-223 as compared to the patients with attenuated responsiveness to DAPT [50]. Interestingly, miR-223-3p in peripheral leukocytes does not correlate with the altered platelet responses in patients treated with clopidogrel [51].

Some reports suggested that miR-125a may have the influence on the development of atherosclerosis [52,53]. Although little is known about its function in the cardiovascular system, it was recently revealed that miR-125a-5p correlates with the number of platelets [54].

As we know, platelet function per se is weaving to such an extent that it makes its usefulness as a disease biomarker for a chronic condition rather weak, as reported in available literature [55]. On the other hand, our findings demonstrate that certain circulating miRNAs are able to reflect the current clinical status of platelet inhibition (significant difference among ASA and clopidogrel) and prognostic power in our study population. Multiple mechanisms influence the outcome of antiplatelet therapy. Clopidogrel is extensively metabolized, and patients’ response to the drug depends on multiple non-genetic and genetic factors [56,57]. The pharmacodynamic response is variable, with up to 40% of patients classed as nonresponders, poor responders or resistant to clopidogrel treatment mainly due to low inhibition of ADP-induced platelet aggregation or activation [58]. This phenomenon should be taken into account while interpreting the data. Our results extend previous findings on the prognostic potential of platelet-derived miRNAs in diabetes [40,46]. The superiority of P2Y_12_ antagonists compared to low-dose ASA deserves further investigation and is in line with recent clinical studies on early ASA interruption in patients on DAPT to prevent bleeding complications [59,60,61,62,63]. Larger studies should be designed to verify the use of miRNAs as potential epigenetic biomarkers with the ability of depicting both genetic and non-genetic risk components.

## 5. Conclusions

In the current study, we demonstrate that in patients with T2DM, higher miR-126 expression is associated with reduced survival in long-term follow-up. Based on the results of the multivariate Cox regression, after correcting for clinical factors, one can conclude that the impact of increased miR-126 expression is unrelated to well-known risk factors for atherothrombotic disease, which further increases the strength of the results. Our findings demonstrate significantly higher expression of miR-126, miR-223 and Let-7e in patients treated with the P2Y_12_ antagonist clopidogrel compared to patients on ASA treatment. Future studies are needed to warrant the clinical usefulness of miR-126 as a mortality risk predictor and to further investigate whether these miRNAs represent only a marker of disease severity or rather a distinct biomarker, which can be modified by pharmacological or lifestyle interventions. Exciting opportunities exist to further pursue platelet miRNAs miR-126, miR-223 and Let-7 to monitor and optimize antiplatelet treatment.

## 6. Study Limitations

The major limitation of the study is that only a fraction of all known miRNAs that are related to platelet reactivity and atherosclerosis pathogenesis were analyzed in this study. The miRNAs analyzed in the present study were chosen a priori based on literature and due to their roles in potential roles in processes related with platelet biology. A more comprehensive genome-wide analysis of miRNAs related to platelet function in T2DM is also warranted. It is therefore possible that other miRNAs might also alter primary or secondary endpoints. In AVOCADO study, three point-of-care tests (CEPI-CT and CADP-CT by PFA-100 and ARU by VerifyNow Aspirin Assay) were applied to evaluate platelet reactivity. Light transmission aggregometry (LTA), which is regarded as the gold standard for platelet reactivity assessment, was not used due to significant time consumption and increased cost of research, which were not included in the original study design [64]. Another limitation is the observational study design with a lack of control group without T2DM; therefore, it is impossible to account for all possible confounding influences. Moreover, due to the limited number of observed clinical endpoints, bias cannot be excluded despite efforts to adjust for baseline differences using multivariate Cox regression analysis. The method of collecting and scrutinizing the follow-up may be to blame for this limitation. Due to the long observational period, personal contact and examination of the patient was difficult. Since it is an observational study that was performed on a population with specific study entry criteria, a small number of events were included in primary or secondary endpoint, and not all clinical data or causes of mortality could be incorporated into the analysis. Because insufficient information is included in the Polish Statistical Registries, we were unable to obtain explicit details on the cause of death of the subject and defined primary end point as all-cause mortality instead of cardiovascular death.

## Figures and Tables

**Figure 1 jcm-10-02371-f001:**
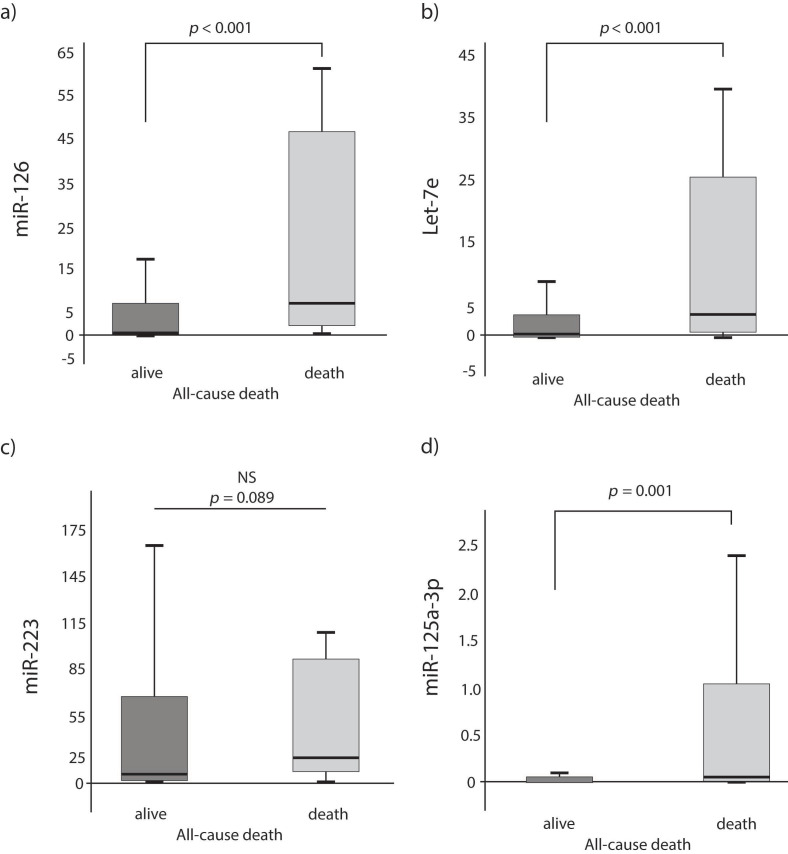
miRNAs expression levels according to survival during the long-term follow-up (**a**) miR-126; (**b**) Let-7e; (**c**) miR-223; (**d**) miR-125a-3p.

**Figure 2 jcm-10-02371-f002:**
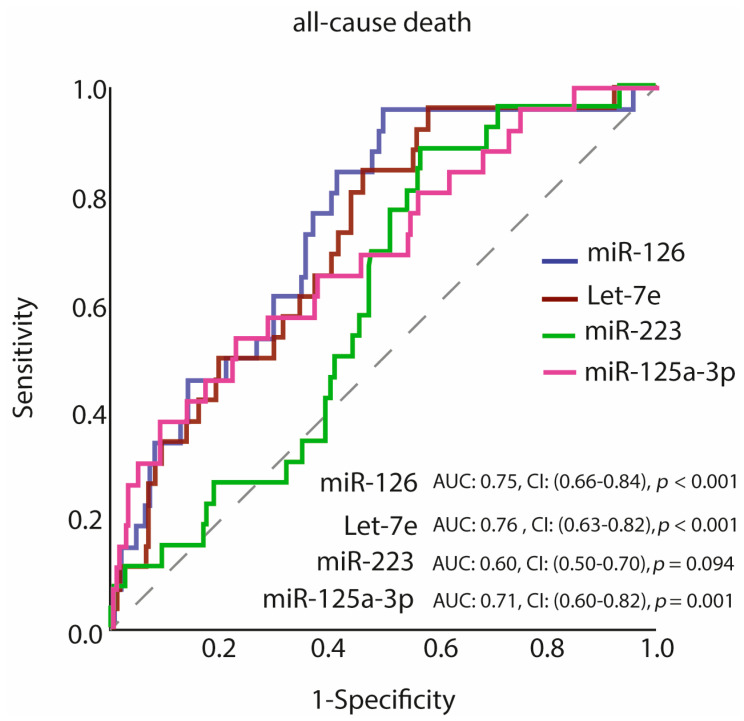
Receiver operating characteristic (ROC) curves of miR-126, Let-7e, miR-223, miR-125a-3p for prediction of all-cause death. Abbreviations: AUC, area under the curve; CI, 95% confidence interval.

**Figure 3 jcm-10-02371-f003:**
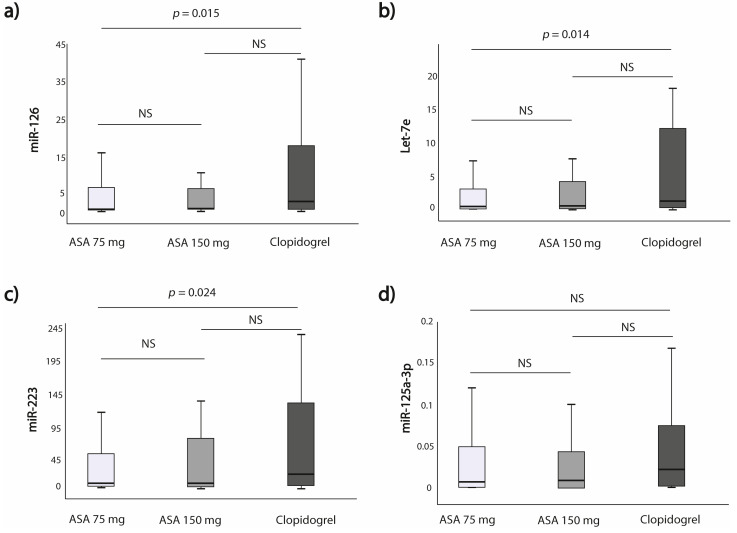
MiRNAs expressions in regard to the type of antiplatelet treatment based on randomization (ASA 75 mg vs. ASA 150 mg vs. Clopidogrel); (**a**) miR-126; (**b**) Let-7e; (**c**) miR-223; (**d**) miR-125a-3p.

**Figure 4 jcm-10-02371-f004:**
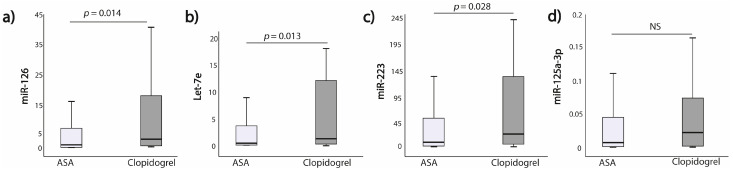
MiRNAs expressions in regard to antiplatelet treatment (ASA vs. Clopidogrel) (**a**) miR-126; (**b**) Let-7e; (**c**) miR-223; (**d**) miR-125a-3p.

**Figure 5 jcm-10-02371-f005:**
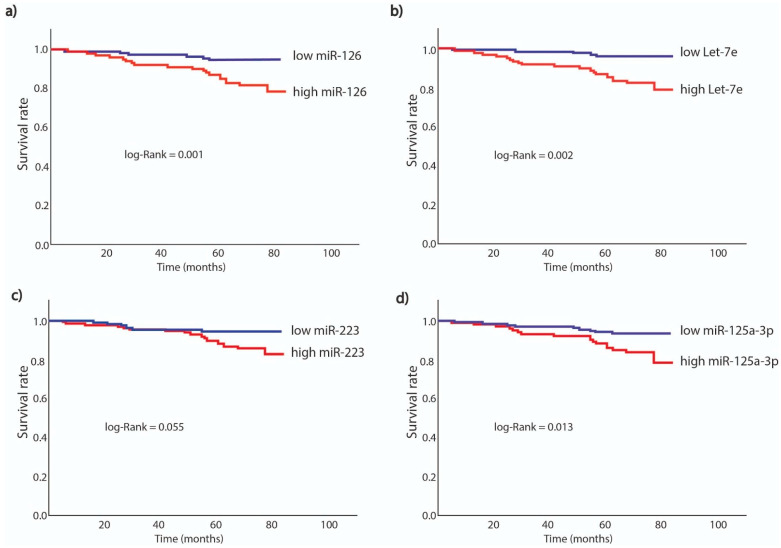
Kaplan–Meier survival analysis for all-cause death. HR are adjusted for clinical variables in a multivariate model with a single miRNA as a covariate (**a**) high vs. low miR-126; (**b**) high vs. low Let-7e; (**c**) high vs. low miR-223; (**d**) high vs. low miR-125a-3p.

**Table 1 jcm-10-02371-t001:** Demographic and clinical characteristics of the study patients.

Patient Demographics	Overall 252 (100%)	Patients Who Died * (N = 26)	Patients Who Survived * (N = 226)	*p*
Age (years) mean ± SD	67.1 ± 8.5	70.52 ± 1.80	66.61 ± 0.60	0.019
Sex (female) n (%)	117 (46%)	111 (49%)	6 (23%)	0.012
Body mass index (BMI)	31.5 ± 12.7	30.21 ± 0.80	31.81 ± 0.99	0.587
Hypertension	228 (92%)	24 (92%)	204 (92%)	1
Dyslipidemia	213 (85%)	20 (77%)	193 (86%)	0.209
HF	91 (37%)	12 (46%)	79 (35%)	0.282
Current smoking	25 (10%)	2 (8%)	23 (11%)	0.745
CAD	136 (55%)	16 (62%)	120 (54%)	0.468
Prior MI	75 (30%)	12 (46%)	63 (28%)	0.058
History of smoking	143 (57%)	20 (77%)	123 (55%)	0.032
Prior ischemic stroke	22 (9%)	5 (19%)	17 (8%)	0.048
Prior TIA	7 (3%)	1 (4%)	6 (3%)	0.708
Laboratory data (mean ± SD)				
White blood cell count (x109/L)	6.9 ± 1.9	6.46 ± 0.40	6.95 ± 0.13	0.222
Platelets (x109/L)	222.6 ± 61.9	204.08 ± 9.62	225.42 ± 4.47	0.113
Hemoglobin (g/dL)	13.7 ± 1.46	13.72 ± 2.67	13.76 ± 0.99	0.969
High-sensitivity C-reactive Protein (mg/dL)	2 [0.3–25.8]	1.9 [1.05–4.0]	2.05 [1.05–4.1]	0.726
Fibrinogen (mg/dL)	402.6 ± 104	407 ± 19.78	404 ± 7.72	0.994
Creatinine (mg/dL)	1 ± 0.31	1.16 ± 0.06	0.97 ± 0.02	0.001
HbA1c	6.5 [6.0–7.4]	6.4 [5.9–7.1]	6.9 [6.3–7.8]	0.060
Tch	158.6 ± 36.5	153.63 ± 6.15	159.34 ± 2.69	0.628
TG	132.4 ± 61.5	123.46 ± 7.13	125.87 ± 3.12	0.990
HDL	50 ± 30.3	47.50 ± 9.31	51.45 ± 2.44	0.634
LDL	83.8 ± 28.3	81.42 ± 5.35	85.36 ± 2.23	0.670
Failure to achieve lipid control				
LDL, %; n **	108 (49%)	13 (54%)	95 (48%)	0.567
HDL, %; n **	100 (44%)	9 (39%)	91 (45%)	0.413
Triglycerides, %; n **	68 (30%)	8 (32%)	60 (29%)	0.778
Concomitant medications n (%)				
ß-blockers	178 (72%)	22 (88%)	156 (70%)	0.054
ACE inhibitors	165 (66%)	14 (56%)	151 (67%)	0.252
Statins	179 (72%)	19 (76%)	160 (71%)	0.630
Calcium channel-blockers	94 (38%)	8 (32%)	86 (38%)	0.532
Proton pump Inhibitors	49 (24%)	8 (31%)	41 (23%)	0.407
Randomization group in the AVOCADO study, %; n				
ASA (75 mg) ***	167 (66%)	11 (42%)	156 (69%)	0.006
ASA (150 mg) ***	32 (13%)	6 (23%)	26 (12%)	0.093
Clopidogrel ***	53 (21%)	9 (35%)	44 (20%)	0.073
ASA total (75 mg + 150 mg) ***	199 (79%)	17 (65%)	182 (81%)	0.073
miRNAs				
miR-126	0.84 [0.16–8.54]	7.23 [1.93–47.3]	0.46 [0.13–6.39]	0.000032
Let-7e	0.64 [0.09–4.10]	3.78 [0.85–25.70]	0.51 [0.08–3.39]	0.000216
miR-223	8.91 [2.09–69.81]	19.98 [6.27–94.85]	8.11 [1.76–67.88]	0.089
miR-125a-3p	0.009 [0.001–0.05]	0.054 [0.006–1.05]	0.0082 [0.001–0.041]	0.0006

* Median follow-up: 5.9 years. ** Failure to achieve lipid control defined as. LDL > 70 mg/dL in patients with a history of coronary artery disease, myocardial infarction, previous stroke or TIA, current smoking or with eGFR < 30 mL/min/1.73 m^2^, and LDL > 100 mg/dL in remaining patients, HDL < 40 mg/dL in men, and < 50 mg/dL in women, triglycerides > 150 mg/dL. *** Study medication (treatment duration: 8 weeks). Data are reported as mean ± standard deviation (SD) and median [interquartile range]. Abbreviations: ACE, angiotensin converting enzyme; ACS, acute coronary syndrome; ASA, acetylsalicylic acid; BMI, body mass index; CAD, coronary artery disease; HDL, high-density lipoprotein; HF, heart failure; LDL, low-density lipoprotein; MI, myocardial infarction; TG, triglycerides.

**Table 2 jcm-10-02371-t002:** Statistical estimates for prediction of all-cause mortality by miR-126, Let-7, miR-223 and miR-125a-3p.

miRNA	c-Index-AUC (95% CI)	*p*	Cut-Off	Sensitivity, %	Specificity, %	Positive Predictive Value, %	Negative Predictive Value, %	Positive Likelihood Ratio
miR-126	0.75 (0.66–0.84)	<0.001	2.078	77%	63%	19%	96%	2.07
Let-7e	0.76 (0.63–0.82)	<0.001	0.8201	81%	66%	18%	96%	1.84
miR-223	0.60 (0.50–0.70)	0.094	6.617	77%	49%	15%	95%	1.49
miR-125a-3p	0.71 (0.60–0.82)	0.001	0.0017	65%	61%	17%	94%	1.78

Abbreviations: AUC—area under the curve; 95% CI—95% confidence interval.

**Table 3 jcm-10-02371-t003:** Univariate and multivariate Cox regression model for prediction of long-term all-cause of mortality for miR-126, Let-7e, miR-223 and miR-125a-3p.

Variable	HR	95% CI		*p*-Value
		Lower	Upper	
High miR-126				
Univariate	4.377	1.749	10.956	0.002
Multivariate *	7.310	2.634	20.284	<0.001
High Let-7e				
Univariate	4.208	1.580	11.206	0.004
Multivariate *	5.845	2.076	16.460	0.001
High miR-223				
Univariate	2.389	0.952	5.977	0.063
Multivariate *	3.073	1.170	8.071	0.023
High miR-125a-3p				
Univariate	2.692	1.198	6.052	0.017
Multivariate *	2.929	1.256	6.828	0.013

* After adjustment for age, gender (male), history of smoking, prior ischemic stroke and eGFR < 30 (mL/min/1.73 m^2^).

**Table 4 jcm-10-02371-t004:** Multivariate Cox regression model including high levels of miRNAs and clinical data.

Variable	HR	95% CI		*p*-Value
		Lower	Upper	
High miR-126	5.821	1.259	24.927	0.024
High Let-7e	3.449	0.578	21.176	0.173
High miR-223	0.367	0.080	1.679	0.196
High miR-125a-3p	1.115	0.408	3.050	0.832
Age	1.068	1.016	1.122	0.009
Gender (male)	4.059	1.235	13.344	0.027
History of smoking	1.656	0.519	5.289	0.395
Prior IS	4.041	1.242	12.646	0.016
eGFR<30	5.879	0.841	41.100	0.074

Abbreviations: HR, hazard ratio; MI, myocardial infarction, 95%CI, 95% confidence interval.

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
