# Peer review of "MiR-126 Is an Independent Predictor of Long-Term All-Cause Mortality in Patients with Type 2 Diabetes Mellitus"

_jcm, 2021, doi:10.3390/jcm10112371_

Round 1
Reviewer 1 Report
The aim of the paper was to determine the ability of several miRNAs to predict mortality and response to anti-platelet treatments among patients with type-2 diabetes melilotus (T2DM). The authors selected patients with diabetes and investigated circulating miRNA value with mortality and in relationship with anti-platelet therapy such as clopidogrel and Aspirin. The data show that the miRNA, miR-26 was increased in patients upon death as compared with patients who survived. In patients treated with aspirin the increased in miR-26 was less pronounced than patients treated with clopidogrel. The authors concluded that miR-26 can be used as a biomarker for survival in diabetes.
The BMI of the patients seems to be very high. How do the authors think this may influence the data? I was also wondering why the patients that were smoking were not excluded.
I understand that the authors do not have a negative control such as T2DM patients with no cardio-vascular disease. If they do, the data should be included in the graphs/table. If not, it would be good for the reader to be informed about it in the introduction or result section, while it is only mentioned in the limitation. It should also be discussed.
Did the authors determine whether there were changes between male and female patients?
Fig 4: the error bars of the group treated with clopidogrel seem very wide so it is hard to see changes. This is in line with the fact that clopidogrel is metabolized differently between individuals.
Is there a change in survival in cardio-vascular patients treated with aspirin and clopidogrel?
The data so far are not enough to suggest that clopidogrel treatment may be less beneficial than ASA. Moreover, the use of the term “changes in miRNA expression” is not verified. For instance, it is a possibility than the expression of miRNA is similar but miRNA is circulating and not captured by cells.
Reviewer 2 Report
In the present report, the Authors determined the ability of several miRNAs to predict mortality and response to antiplatelet treatment among T2DM patients. They recruited 252 diabetics receiving either acetylsalicylic acid (ASA) 75mg, ASA 150 mg or clopidogrel. They found that miR-126 expression was a strong and independent predictor of long-term all-cause mortality among patients with T2DM. Moreover, miR-223, miR-126 and Let-7e present significant interactions with antiplatelet treatment regimens and clinical outcomes. Minor comments are listed below:
- In the study design section, there was mentioned that patients who did not exhibit high platelet reactivity on ASA 75 mg daily continued therapy, and patients with high platelet reactivity were randomized to ASA 150 mg or clopidogrel 75 mg. An important element limiting the effectiveness of antiplatelet therapy is the occurrence of low response to antiplatelet drugs. Multiple mechanisms may be responsible for low antiplatelet response, including clinical, cellular, and genetic factors. Upregulation of multiple platelet adhesion, activation and aggregation pathways, and the release of inflammatory markers and prothrombotic factors appear to be of particular importance. Before treatment, high baseline platelet reactivity may also contribute to decreased antiplatelet effects induced by clopidogrel or aspirin, especially in patients with acute coronary syndromes and diabetes. Probably this phenomenon occurred in some of the patients. Could it have influenced the obtained results?
- It is difficult to point out that the presented findings demonstrate that certain circulating miRNAs are able to reflect the current clinical status of platelet inhibition based solely on the significant difference between ASA and clopidogrel. No tests were performed to assess platelet activation (for example, impedance aggregometry).
Round 2
Reviewer 1 Report
The authors properly addressed all my concerns. Good luck with your future work!